# Membrane Contact Sites in Yeast: Control Hubs of Sphingolipid Homeostasis

**DOI:** 10.3390/membranes11120971

**Published:** 2021-12-09

**Authors:** Philipp Schlarmann, Atsuko Ikeda, Kouichi Funato

**Affiliations:** Graduate School of Integrated Sciences for Life, Hiroshima University, Kagamiyama 1-4-4, Higashi-Hiroshima 739-8528, Japan; phschlarmann@gmail.com (P.S.); atsukoikeda@hiroshima-u.ac.jp (A.I.)

**Keywords:** sphingolipids, membrane contact sites, metabolism, ceramides, non-vesicular transport, lipotoxicity, yeast, *Saccharomyces cerevisiae*

## Abstract

Sphingolipids are the most diverse class of membrane lipids, in terms of their structure and function. Structurally simple sphingolipid precursors, such as ceramides, act as intracellular signaling molecules in various processes, including apoptosis, whereas mature and complex forms of sphingolipids are important structural components of the plasma membrane. Supplying complex sphingolipids to the plasma membrane, according to need, while keeping pro-apoptotic ceramides in check is an intricate task for the cell and requires mechanisms that tightly control sphingolipid synthesis, breakdown, and storage. As each of these processes takes place in different organelles, recent studies, using the budding yeast *Saccharomyces cerevisiae*, have investigated the role of membrane contact sites as hubs that integrate inter-organellar sphingolipid transport and regulation. In this review, we provide a detailed overview of the findings of these studies and put them into the context of established regulatory mechanisms of sphingolipid homeostasis. We have focused on the role of membrane contact sites in sphingolipid metabolism and ceramide transport, as well as the mechanisms that prevent toxic ceramide accumulation.

## 1. Introduction

Sphingolipids are important components of cellular membranes that exert various functions, depending on their structural maturation and subcellular localization. Sphingolipids are synthesized in the endoplasmic reticulum (ER) and transported to the Golgi, where they are equipped with a variable headgroup. The resulting products are termed complex sphingolipids and are integrated mainly into the plasma membrane (PM), where they contribute to the organelle’s barrier function and identity. In addition, PM sphingolipids assemble with sterols and membrane proteins into microdomains that drive many PM-related processes, such as endocytosis, nutrient transport, and pH maintenance, by limiting lateral movement and increasing the interaction of functional partners [1]. In contrast, simple sphingolipid precursors, such as long-chain bases (LCBs) and ceramides, are not only necessary intermediates for the synthesis of complex sphingolipids but also function as signaling lipids in processes, including cell migration, stress response, senescence, differentiation, and endocytosis [2]. Of special interest are the opposing roles of sphingolipid precursors in determining cell fate for ceramide with pro-apoptotic properties and LCB-1-phosphate anti-apoptotic properties [3,4]. Due to this fact, ceramide has attracted increasing attention as a tumor suppressor lipid, and the mechanisms that regulate ceramide levels have become the subject of intense investigation [5]. The budding yeast, *Saccharomyces cerevisiae*, has been employed as the model organism of choice to investigate sphingolipid homeostasis, and studies have revealed an extensive kinase network evolving around the target of rapamycin complexes 1 (TORC1) and 2 (TORC2), which senses nutrient levels at the vacuole, as well as lipid homeostasis at the PM, and adjusts sphingolipid synthesis accordingly [6,7,8,9]. Other mechanisms contributing to sphingolipid homeostasis are sphingolipid breakdown, which takes place in the vacuole, ER, and mitochondria, as well as storage of sphingolipids as acylceramide in lipid droplets (LDs) [10,11,12]. As these processes involve a variety of organelles, membrane contact sites (MCSs) have been proposed to mediate non-vesicular interorganellar transport of sphingolipids and integrate regulatory mechanisms for sphingolipid homeostasis. In this review, we summarize the metabolic and catabolic pathways of sphingolipids and the regulatory mechanisms that control them, specifically MCSs and non-vesicular ceramide transport. We have focused on studies using *S. cerevisiae* but also highlight research on mammalian cell lines, when complementation is needed.

## 2. Sphingolipid Synthesis, Catabolism, and Trafficking in Yeast

In *S. cerevisiae*, the major pathways and enzymes involved in sphingolipid synthesis have already been identified (Figure 1A). Sphingolipid biosynthesis begins with the condensation of cytosolic serine and palmitoyl-CoA by the ER-resident serine palmitoyltransferase (SPT) complex, yielding 3-ketodihydrosphingosine (KDS). The SPT complex contains three subunits: Lcb1, Lcb2, and Tsc3. Because its active site is at the interface of the two catalytic subunits, Lcb1 and Lcb2, facing the cytosol, it is suspected that KDS is incorporated into the outer leaflet of the ER membrane after its synthesis [13,14,15]. Subsequently, KDS is reduced to dihydrosphingosine (DHS) or, in a following step, hydroxylated to phytosphingosine (PHS), by enzymes that also face the outer leaflet of the ER [16]. DHS and PHS are termed “LCBs,”, which act as the backbone of sphingolipids. Both LCB-types can be converted with a CoA-activated very long-chain fatty acid (VLCFA) into dihydroceramide and phytoceramide, respectively, through ceramide synthase. Ceramide synthase is a heteromeric protein complex, containing three subunits, Lag1, Lac1, and Lip1. Lag1 and Lac1 are homologous proteins that feature eight transmembrane domains (TMDs) and a conserved Lag motif, which extends over the 5th and 6th TMDs and presumably harbor the active site of the proteins [17]. Both proteins are functionally similar, but differ in their substrate specificity, with Lag1 being better than Lac1 in converting PHS to phytoceramide [18]. This difference already accounts for the distinct roles of Lag1 and Lac1 in aging and further suggests that each LCB isoform and its derivatives have unique physiological functions [18]. The third subunit of the ceramide synthase Lip1 features a single TMD and exerts its essential function in the complex, through the transmembrane or luminal part of the protein [19]. The topology of ceramide synthase is not understood in detail. However, a model was proposed, in which the catalytic center of the ceramide synthase forms a channel that connects the leaflets of the ER membrane [17]. Translocation between the ER leaflets is essential for both the substrate and product of ceramide synthase. For example, after a series of phosphorylation and dephosphorylation steps, exogenously added DHS is delivered into the inner layer of the ER, where it is used for ceramide synthesis [20,21]. Therefore, translocation is necessary to ensure the conversion of endogenously synthesized and exogenously added DHS. Likewise, ceramide is found on both sides of the ER membrane and can be transported by vesicular transport from either side to the next destination of the sphingolipid synthesis pathway, the Golgi. Ceramides present on the cytosolic side are also available for delivery to the Golgi via non-vesicular transport.

The transport of ceramide from the ER to the Golgi depends mainly on the COPII vesicles of the secretory pathway, similar to that of proteins. Glycosylphosphatidylinositol (GPI)-anchored proteins are also synthesized in the ER and transported to the Golgi apparatus by COPII vesicles [24]. Ceramides in the inner layer of the ER are involved in remodeling GPI lipid moieties and seem to coalesce with GPI-anchored proteins during vesicular transport (Figure 1A), as defects in GPI-anchored biosynthesis affect vesicular transport of ceramide [25,26,27]. Conversely, normal ceramide synthesis is required for GPI-anchored protein transport [28]. GPI-anchored proteins are segregated into lipid microdomains in the ER and are sorted into specific transport vesicles [29]. Recent observations, using high-speed, super-resolution live imaging microscopy, have confirmed this model and suggest that the length of ceramide in the raft-like structure is important for the sorting mechanism [30]. Although vesicular transport is the main pathway for ceramide to reach the Golgi, alternative routes exist that maintain some transport even under conditions that block the secretory pathway [31]. This so-called non-vesicular ceramide transport depends on tethering proteins at ER-Golgi contact sites, which facilitate the transfer of ceramide between the surface of the ER and Golgi, which will be discussed later in the review.

Ceramide is converted to complex sphingolipids in the Golgi apparatus. In *S. cerevisiae*, the major complex sphingolipid is inositol phosphorylceramide (IPC), which is synthesized by IPC synthase via the transfer of an inositol phosphate headgroup from phosphatidylinositol to ceramide. IPC synthase is a protein complex with two known membrane-spanning subunits, Aur1 and Kei1. While Aur1 contains the active site of the complex that faces the lumen of the Golgi [32], Kei1 was shown to be essential for the localization of the complex and its catalytic ability [33]. Because ceramides transported by the non-vesicular pathway are incorporated into the outer layer of the Golgi and used for IPC synthesis [31], it is apparent that a yet unidentified mechanism translocates ceramides to the luminal site. After its synthesis, IPC can be further modified by the mannosylation of its headgroup to form mannose inositolphosphorylceramide (MIPC) or mannose di (inositolphosphoryl) ceramide (M(IP)_2_C). The synthesis of MIPC depends on the Csh1p/Csg1p/Csg2p complex, and the active site of Csg1p is also predicted in the lumen of the Golgi [34]. Notably, the catalytic subunits of the complex show a different affinity to the dihydro- and phyto-isoforms of IPC, such as the previously mentioned ceramide synthase subunits [35]. Conversion of MIPC to M(IP)_2_C is carried out by Ipt1 [35,36], and complex sphingolipids are transported to the PM. This transport is once more coupled to vesicular transport, as it was shown that the transport of IPC, MIPC, and M(IP)_2_C to the PM almost completely stops when vesicular transport is blocked [22,37].

The remodeling of membranes, in response to cell cycle progression and environmental changes, requires enzymes that can reverse each step of the sphingolipid synthesis pathway. Inositol-phosphosphingolipid phospholipase C (Isc1) localizes to the ER and mitochondrial outer membrane and cleaves the polar head group from complex sphingolipids at the C1 position, yielding phytoceramide and a free head group [38,39] (Figure 1A and Figure 2C). ER and endosomal enzymes degrade phytoceramide in multiple steps, further into PHS and 2-hydroxyhexadecanal, which can be recycled into glycerophospholipid metabolism [40,41,42]. Defects in sphingolipid degradation and recycling are connected to a variety of human lysosomal storage diseases such as Niemann Pick type C, a fatal neurodegenerative disorder, caused by mutations in the *NPC1* or *NPC2* gene, which is characterized by lysosomal accumulation of cholesterol and sphingolipids [43]. In yeast, sphingolipid breakdown is also connected to endosomal and vacuolar degradation because deletion of Isc1 leads to defective lysosomal trafficking and vacuolar function [44,45]. Cells lacking Isc1 not only build up complex sphingolipids but also some sphingolipid precursors, including very long-chain ceramide species that are presumably derived from increased de novo synthesis [46]. Increased levels of long-chain ceramides activate the catalytic subunit of the protein phosphatase type 2A (PP2A) complex Sit4, which links ceramide and vacuolar homeostasis by negatively regulating protein sorting, vesicular trafficking, vacuole function, and autophagy [44]. Consequently, double deletion of Sit4 and Isc1 restores vacuole function [44]. Sphingolipids not only control endosomal maturation via Sit4 but are also sorted and recycled themselves on endosomes. Ncr1/2, yeast orthologs to the human Npc1/2 proteins, were suggested to sort sphingolipids on endosomes with IPC being directed to the vacuolar membrane, and MIPC and M(IP)2C being returned to the Golgi and subsequently transported to the PM [47]. The recycling of PM sphingolipids depends on the Golgi-associated retrograde protein (GARP) complex, which tethers retrograde transport vesicles derived from endosomes to the late Golgi [48]. Blocking this pathway induces toxic accumulation of LCBs, which suggests increased degradation of complex sphingolipids in the vacuole [49]. In addition to sphingolipid sorting, Ncr1/2 plays a role in sterol transport from endosomes to the vacuolar membrane to form sterol-and sphingolipid-enriched raft-like domains that mediate microautophagy [50]. Because GARP deficiency also alters sterol metabolism [49], it seems likely that sphingolipids and sterols are sorted via the same Ncr1/2-mediated mechanism.

## 3. Regulation of Sphingolipid Metabolism

LCBs, ceramides, and other intermediates in sphingolipid metabolism are vital but potentially toxic signaling molecules [2,3]. Supplying complex sphingolipids to the PM based on need and simultaneously preventing the toxic accumulation of metabolites is an intricate task for cells. It is, therefore, not surprising that many partially interconnected pathways exist that respond to environmental changes and regulate different steps along the sphingolipid synthesis pathway (Figure 1B). The formation of LCBs from serine and palmitoyl-CoA by SPT is the rate-limiting step of sphingolipid synthesis and is the main target of regulation. It was shown that LCB synthesis depends on the external uptake of serine, indicating that sphingolipid metabolism is not regulated prior by de novo serine production [51]. The catalytic activity of the SPT complex mainly depends on the presence of two regulatory proteins, Orm1 and Orm2, which inhibit SPT when they are associated with it [52,53]. Orm1/2 are downstream effector proteins of an extensive kinase network that coordinates sphingolipid metabolism with other cellular tasks, such as cell cycle progression, nutrient uptake, and stress response. Here, we break down the pathways of this kinase network targeting Orm1/2 starting with the initial stimuli:

Sphingolipid deficiency: PM stress caused by sphingolipid deficiency is accompanied by changes in membrane properties and rearrangement of lateral microdomains. Slm1, Slm2, and Nce102 are proteins that localize at steady state to the eisosome, a protein complex that is part of the static PM microdomain, known as the membrane compartment which is occupied by Can1 (MCC) (Figure 1B). Slm1/2 are proteins that localize to the PM by binding to phosphatidylinositol 4,5-bisphosphate (PI4,5P2), whereas Nce102 is an integral PM protein. Depletion of sphingolipids disrupts the lipid and protein organization of the MCC, causing Slm1/2 and Nce102 to dissociate from the eisosome within the membrane plane. In this way, Slm1/2 become available to bind and activate the PM resident TORC2, which in turn phosphorylates and activates the cytosolic protein kinase Ypk1 [7,54]. Similar to Slm1/2, dissociation of Nce102 alleviates its inhibitory effect on the eisosome resident protein kinases Pkh1 and Pkh2, which in turn phosphorylate Ypk1 as well, but at different sites compared to TORC2-mediated phosphorylation [55]. Ypk1 phosphorylates several targets, including Orm1/2, triggering their release from the SPT complex and stopping SPT inhibition [7,53,56]. Phosphorylation of Orm1/2 not only causes the proteins to dissociate from the SPT complex but leads to their degradation, via the recently identified endosome and Golgi-associated degradation (EGAD) pathway [57]. In this pathway, phosphorylated Orm2 is exported to the Golgi and endosomes, where it is selectively polyubiquitinated by the Dsc ubiquitin ligase complex, extracted from the membrane by the unfoldase Cdc48, and finally degraded by the proteasome [57]. Degradation of already inactivated Orm2 is necessary because accumulation of Orm2 can divert Ypk1 activity away from other important targets, such as ceramide synthase [58]. Unlike the SPT complex, which depends on Orm1/2 interaction, ceramide synthase and many other key enzymes in sphingolipid biosynthesis are regulated by direct phosphorylation through protein kinases. Both the ceramide synthase subunits Lag1 and Lac1 are phosphorylated by Ypk1 to activate ceramide synthesis [58,59]. Consequently, TORC2/Ypk1 action controls the first two consecutive steps of sphingolipid synthesis, which ensures a coherent flux of metabolites [58] and provides a feedback mechanism, in response to sphingolipid depletion [7]. Moreover, Ypk1 was suggested to further regulate complex sphingolipid synthases such as Csh1 and Sur1 as these enzymes are phosphorylated at Ypk1 typical phosphorylation motifs, in response to myriocin-induced SPT inhibition [60].

Cell cycle progression and growth: The cell cycle checkpoint kinase Swe1 induces Orm1/2 phosphorylation and LCB synthesis independently of Ypk1 [61]. Swe1 is activated and halts cell cycle progression in response to defective lipolysis, which is a source of lipid precursors for sphingolipid synthesis [62]. Therefore, Orm1/2 phosphorylation by Swe1 could be a feedback loop to maintain a steady supply of LCBs. Moreover, casein kinase 2 (CK2), a major regulator of various pro-growth cellular events and suppressor of apoptosis [63,64], promotes ceramide synthesis by directly phosphorylating the *C-*terminal cytoplasmic domains of the ceramide synthase subunits Lag1 and Lac1 [65]. CK2 dependent phosphorylation of ceramide synthase was suggested to maintain its localization to the ER membrane via the COP I-dependent C-terminal dilysine ER retrieval pathway [65].

Starvation: Orm1/2 are also phosphorylated via the TORC1 pathway (Figure 1B), which senses amino acids at the vacuole and adjusts cellular metabolism in response to amino acid starvation [66]. TORC1 is inhibited under these conditions and is unable to phosphorylate a variety of downstream targets, including the cytosolic protein phosphatase type 2A (PP2A), which is suppressed by TORC1 at steady state [67]. PP2A is responsible for dephosphorylation and activation of the protein kinase Npr1, which in turn phosphorylates Orm1/2 but at different sites compared to Ypk1 [6]. In contrast, Npr1 mediated Orm1/2 phosphorylation does not affect LCB synthesis, but instead activates Orm1/2 to promote the synthesis of complex sphingolipids downstream of the SPT complex through an unidentified mechanism [6]. Because increased complex sphingolipid levels at the PM facilitate the integration of amino acid permeases, TORC1 inhibition provides a feedback loop to promote nutrient uptake during starvation [6]. Nevertheless, it was shown that TORC1 controls the early steps of sphingolipid metabolism by mechanisms other than Orm1/2 phosphorylation. The introduced PP2A branch of TORC1 was proposed to control the phosphorylation and activity of the VLCFA elongase Elo2 via Mck1 kinase [68]. The details of this pathway remain unclear, and contradicting reports have been published on whether Mck1 mediated phosphorylation of Elo2 promotes [68] or inhibits Elo2 activity [69]. Furthermore, steady-state levels of sphingolipids are regulated through the Sch9 effector branch of TORC1, which represses the expression of the ceramidase genes YDC1 and YPC1 as well as ceramide synthase genes LAG1 and LAC1 [70]. Conversely, Sch9 seems to mediate hydrolytic ceramide production during diauxic shift, as Sch9 is essential for the proper translocation of inositol phosphospingolipid phospholipase C, Isc1, from the ER to the mitochondria [70]. Whereas the main role of TORC1 is to adjust cellular metabolism to amino acid availability, other stressors regulate TORC1 function as well such as carbon or phosphorous starvation, hyperosmolarity, oxidants and heat [71,72].

Heat stress and cell wall integrity (CWI) defects: Heat stress induces TORC1 sequestration into stress granules, thereby inhibiting TORC1 function in the vacuole and activating sphingolipid synthesis in a way that is similar to starvation [73]. In addition, elevated temperature is another cause of PM stress that induces sphingolipid synthesis by promoting phosphoinositide-signaling at the PM and subsequently activates two parallel responses, the previously introduced TORC2 pathway and the CWI pathway. Phosphatidylinositides give organellar membranes characteristic properties and drive many membrane-related processes by recruiting effector proteins that specifically recognize their phosphorylation patterns [74]. The two characteristic phosphoinositides of the PM are phosphatidylinositol 4-phosphate (PI4P) and PI(4,5)P2, which are synthesized by the subsequent kinase activities of Stt4 and Mss4, respectively. PM phosphoinositides and sphingolipids seem to be mutually regulated because the localization and activity of Mss4 depends on sphingolipid synthesis [75,76]. Conversely, it was shown that the substrate of Mss4, PI4P, accumulates at the PM upon inhibition of sphingolipid synthesis [77]. Heat shock induces a transient increase in PM phosphoinositides, which recruit PI(4,5)P2-binding Slm1/2 proteins as well as Pkh1 to the PM [78,79] and promote the Pkh-dependent phosphorylation of Slm1/2 [80]. Thus, sphingolipid synthesis is activated via the Pkh1-TORC2-Ypk1 pathway. Simultaneously, elevated levels of phosphoinositides recruit the guanine nucleotide exchange factors Rom1 and 2 to the PM [79], which together with cell-surface sensors Wsc1, 2 and 3 stimulate nucleotide exchange on the small G protein, Rho1, the designated master regulator of the CWI pathway [81]. The CWI pathway controls several stress response mechanisms, including the high osmolarity response pathway, which alleviates the accumulation of aberrant reactive oxygen species caused by heat stress or inhibited sphingolipid synthesis [82,83,84]. In contrast, it was shown that the VLCFA elongase Elo2 is phosphorylated and inactivated by Mck1 kinase, a presumed downstream effector of the CWI pathway (Figure 1B), and a model was proposed in which CWI signaling is inactivated in response to perturbed sphingolipid synthesis to de-repress Elo2 [69]. However, this model requires additional investigation, as it remains unclear how signals from the PM are transduced to Elo2 and how Elo2 inhibition is coordinated with other CWI functions.

Heat induced sphingolipid synthesis last approximately 30 min after which LCBs reach near-basal levels [36,85]. Timely monitoring of sphingolipid synthesis and Orm2 phosphorylation in response to heat stress revealed that Orm2 phosphorylation decreases when LCBs reach peak levels, while ceramide synthesis peaks 10 min later [86]. As this effect is completely diminished by inhibition of SPT and accelerated by exogenously adding LCBs it was proposed that LCBs activate protein phosphatases that act upstream of Orm1/2 to restore SPT inhibition after the heat response. The protein phosphatase PP2A, which is also activated by ceramides [44], has been proposed to mediate inhibition of Ypk1 [86] or Pkh1/2 [87].

## 4. MCSs in Sphingolipid Metabolism

Cellular organelles maintain several domains that are in juxtaposition with other organelles. In recent years, it has become apparent that these contact sites perform a variety of functions ranging from lipid synthesis and transfer to intracellular processes such as autophagosome formation and membrane fission [88,89]. As sphingolipid metabolism extends over most organelles, the role of these contact sites in sphingolipid transport and homeostasis has been closely investigated. Here, we provide an overview of how the MCSs are directly involved in sphingolipid metabolism.

Contact sites between the cortical ER and PM cover about 45% of the inner layer of the PM [90,91,92] and are essential for maintaining PI4P and Ca^2+^ signaling pathways that control ceramide synthesis (Figure 2A). ER-PM contacts are established by specialized ER-resident transmembrane proteins called tethers, which feature large disordered cytosolic regions that attach to the opposing PM through lipid or protein interactions. Seven tethering proteins have been identified at this site: three tricalbins (Tcb1-3), Ist2, Scs2, Scs22 [90], and Ice2 [93]. Cells show an almost complete loss in cortical ER-PM association when tethers are deleted, which causes reduced growth and perturbations in lipid homeostasis, including accumulation of PI4P at the PM [94], disorganization of PM-sterols [93], and overall reduction in cellular complex sphingolipid levels [90]. Until recently, a favored model was that these effects are accounted for by dysfunctional proteins that localize to the cortical ER and regulate PM-lipid homeostasis in *trans*. Osh proteins, a conserved family of lipid exchange proteins related to mammalian oxysterol-binding proteins, are the most promising candidates for regulating lipid homeostasis via ER-PM contact sites. Osh proteins directly interact with ER-PM tethers [95], and deletion of Osh proteins causes phenotypes, similar to the deletion of ER-PM tethers, such as PI4P accumulation [93]. Osh5 and Osh8 were proposed to transport phosphatidylserine (PS) to the PM, in exchange with PI4P [96], which is degraded in the ER by the PI4P phosphatase Sac1, thus maintaining a PI4P concentration gradient that powers the lipid exchange [97]. Osh3 functions as a sensor of PM PI4P and can activate Sac1 to act in *trans* on the PM [95]. In addition, Osh4 was shown to exchange sterols for PI4P between lipid bilayers [98,99], although it is debated whether this transport occurs at the ER-PM or only at ER-Golgi contacts [93] (Figure 2B). Because of their ability to maintain PM levels of PS, PI4P, and possibly sterols, Osh proteins establish a nanoscale membrane lipid environment that promotes PI4P 5-kinase activity and PI (4,5) P synthesis at the PM [94]. Whereas PI (4,5) P synthesis is important for recruitment of Slm1/2 to the PM and TORC2-Ypk1 signaling, it was found that deletion of ER-PM tethers reduced Ypk1 phosphorylation, following heat stress, by only 30% and did not affect Pkh1 and TORC2 localization to the PM [100]. Therefore, disruption of Osh function in cells without ER-PM tethering likely accounts for PI4P accumulation and disorganization of PM-sterols but cannot explain the overall reduction in complex sphingolipids.

Instead, ER-PM contacts have been suggested to modulate intracellular Ca^2+^ and calcineurin signaling, which acts antagonistically to TORC2 signaling (Figure 2A). As previously described, the Pkh1-TORC2-Ypk1 pathway is activated in response to heat stress to promote the first two steps of sphingolipid synthesis. In this event, Ypk1 phosphorylates Orm1/2 to de-repress SPT and phosphorylates Lag1 and Lac1 to activate ceramide synthesis [58]. Simultaneously, heat stress induces Ca^2+^ influx and Ca^2+^/calmodulin-mediated activation of the protein phosphatase calcineurin, which reverses Lag1 and Lac1 phosphorylation [58,59]. In addition, calcineurin activates *ORM2* transcription [8] and represses Orm1/2 degradation via the EGAD pathway by promoting dephosphorylation of Orm1/2 either directly or indirectly [101]. The opposite roles of TORC2 and calcineurin were further demonstrated by the fact that deletion of the calcineurin regulatory subunit B restored phytoceramide levels in cells containing a temperature-sensitive allele of the TORC2 subunit *TSC3* [59]. TORC2 is a major negative regulator of endocytosis [102,103] in contrast to Ca^2+^ and calcineurin, which promote endocytosis universally in neuronal and non-neuronal secretory cells [104] and are suspected to act in the same manner in yeast. Because endocytosis takes place on cortical ER-free PM zones [105], it was speculated that by extension of dynamic ER-PM contacts, Ca^2+^ and calcineurin activity could be contained [100]. This is supported by the fact that cells with deleted ER-PM tethering proteins showed increased cytoplasmic Ca^2+^ and calcineurin activity [100]. Among the ER-PM tethering proteins the tricalbins seem to be possible candidates to drive this process since they have been suggested before as calcium effector proteins. Tricalbins feature several C2 calcium-binding domains [90] and possess cross membrane phospholipid transfer capabilities that are increased in the presence of calcium [106]. In addition, cells lacking Tcb1/2/3 display only a small reduction in ER-PM association but show strong defects in PM integrity following heat shock, which induces calcium signaling [90]. A very recent study showed that, upon stress induction, tricalbins promote non-vesicular phosphatidylserine and phosphatidylethanolamine transport from the ER to the PM [107]. Moreover, Tcb3 co-localizes upon heat stress with the PM protein Sfk1 which is important for PM phospholipid homeostasis and tricalbins mediate the heat-induced recruitment of the CWI factor Pkh1 to the PM [107]. Taken together, ER-PM tethering proteins seem to serve not only as physical membrane bridges, which promote the action of effector proteins acting in *trans* but contribute themselves as effectors and mediators to the TORC2/Ca^2+^ rheostat and CWI pathway which control sphingolipid homeostasis.

Despite their evident role in maintaining lipid homeostasis, the loss of ER-PM contact sites does not affect cellular growth. A study on synthetic lethal mutations in a strain lacking ER-PM contact sites identified components of the endosomal sorting complex required for transport (ESCRT) III complex as essential mediators of lipid homeostasis when tethering fails [108]. The ESCRT machinery was originally known to drive the formation of multivesicular bodies for endosomal degradation of integral membrane proteins [109]. In this pathway, ESCRT III filaments are nucleated by earlier ESCRT complexes at cargo-enriched sites to form a scaffold that invaginates the membrane and buds vesicles into the lumen of the endosome. The vesicles are then released by Vps4 mediated membrane scission accompanied by ATP consumption. Later studies reported that the ESCRT machinery is employed for membrane remodeling at other sides as well [110]. Schmidt et al. showed that the ESCRT III complex and Vps4 are recruited to the PM to maintain membrane integrity, in response to reduced membrane tension and TORC2 inhibition [101]. Conversely, deletion of Vps4 causes hypersensitivity to PM stress, deregulated calcineurin signaling, and activation of TORC2 signaling as a counter measure to calcineurin-inhibited sphingolipid synthesis [101]. Therefore, such as ER-PM contact sites, ESCRT III/Vps4 acts as a mediator of the TORC2/Ca^2+^ rheostat and possibly provides a back-up mechanism when ER-PM contacts are deleted. However, the details of this process, especially the role of Vps4, remain elusive, since the deletion of Vps4 in cells lacking ER-PM tethering did not cause synthetic lethality, unlike deletion of ESCRTIII components [108].

The loss of ER-PM contacts under certain conditions is also compensated by vacuole-PM contact site formation. It was recently discovered that the PM of living yeast cells forms micron-scale protein-depleted regions termed “void zones” [111]. These domains resemble the artificial, phase-separated regions seen in giant unilamellar vesicles and giant PM vesicles but are in fact maintained by cells in an energy-dependent manner. Their physiological function is yet unknown, and void zones have been observed to date only under very specific conditions, such as in PS-deficient yeast cells grown at high temperatures. Void zones lack association with the cortical ER but instead form contact sites with the vacuole, possibly to prevent membrane integrity defects at the interface of the phase separation which is prone to high membrane permeability [112]. Because the formation of void zones at the PM depends on sphingolipid synthesis [111], void zone-vacuole contacts may be involved in regulating sphingolipid metabolism or organization of the PM.

## 5. MCSs in Non-Vesicular Transport of Ceramide

The presence of a non-vesicular ceramide transport mechanism between the ER and Golgi apparatus was first proposed since complex sphingolipid synthesis in the Golgi was maintained at a low level, even after complete block of vesicular trafficking [31]. Both fluorescence and electron microscopic studies have shown that 10–30% of medial Golgi compartments containing the IPC synthase Aur1 are closely associated with the ER under non-stress conditions in wild-type yeast cells [113]. During ER stress, the ER-medial Golgi contacts increase, suggesting that the ER-Golgi contact is tightly regulated in response to environmental cues. The increase in ER-medial Golgi contact sites involves the tethering protein Nvj2, which is mainly localized at the nuclear envelope–vacuole junction (NVJ) under non-stress conditions but re-localizes to ER-medial Golgi contacts, following ER stress or ceramide overproduction (Figure 2B and Figure 3A). Liu et al. demonstrated that Nvj2 promotes the non-vesicular transfer of ceramides from the ER to the Golgi complex. As an artificial ER-Golgi tether is unable to completely restore ceramide transfer in cells lacking Nvj2, Njv2 may directly facilitate ceramide transport, in addition to functioning as a tether. Although its direct role remains unclear, it has been proposed that the synaptotagmin-like, mitochondrial (SMP) domain of Nvj2 transfers ceramide from the ER to the Golgi, because a mutation in the SMP domain that probably prevents lipid binding fails to facilitate IPC synthesis via non-vesicular ceramide transport [113]. In addition, Nvj2 features a pleckstrin homology (PH) domain. It is likely that the PH domain is necessary for Nvj2 to bind Golgi membranes; although, which lipids and/or proteins are bound and how the localization of Nvj2 is regulated by ER stress remains unknown.

The members of the yeast tricalbin protein family Tcb1, Tcb2, and Tcb3 are orthologs of the mammalian extended synaptotagmin proteins and henceforth also feature an SMP domain. The tricalbins were identified by Manford et al. [90] as tethering proteins that are highly conserved throughout eukaryotes and are localized to the ER where they form contacts with the PM. In a recent study, we found that Tcb3 is recruited to ER-Golgi contact sites during ER stress or when overexpressed [12] (Figure 2B and Figure 3A). Tcb3 preferentially associates with contact sites between the ER and medial Golgi, where Aur1 is localized, compared to contact sites between the ER and cis Golgi or between the ER and trans-Golgi. Importantly, deletion of tricalbins perturbs IPC synthesis via non-vesicular ceramide transport without affecting ceramide synthesis and IPC synthase activity, implying that tricalbins are involved in non-vesicular ceramide transport. Both the SMP and C2 domains of Tcb3 are required for IPC synthesis, whereas only the C2 domains contribute to the formation of MCS between the ER and medial Golgi. Therefore, similar to the SMP domain of Nvj2, the SMP domains of tricalbins may directly transfer ceramide at ER-medial Golgi contacts.

The deletion of tricalbins results in the accumulation of acylceramide, which is generated from ceramide by O-acyltransferases, Dga1 and Lro1 [12]. Conversion of ceramide into acylceramide is thought as a response to access ceramide accumulation caused by the defect of non-vesicular ceramide transport. As acylceramides are synthesized in the ER and stored with other neutral lipids in LDs (Figure 3A), it is possible that acylceramide stored in the LD may be directly transported to the Golgi via LD-Golgi contact sites, and tricalbins may facilitate the delivery of acylceramide from the LD. Contradicting this model is, however, the fact that isolated ER- and Golgi-enriched membrane fractions can reconstitute non-vesicular ceramide transport process *in vitro* [31]. It is also possible that acylceramides are transiently present in the ER and are delivered to the Golgi apparatus via the ER-Golgi contact sites. Furthermore, under non-stress conditions, tricalbins and Nvj2 localize to the ER-PM contact and NVJ, respectively [90,114]. Thus, it cannot be ruled out that ceramides are transported from the ER indirectly to the Golgi apparatus via the PM or vacuole. In this regard, it is noteworthy that a fraction of Golgi compartments associate with the cortical ER that lies near the PM [12] or vacuole [115]. Mutations in Mdm1, an NVJ tethering protein, induce hypersensitivity to a specific sphingolipid synthesis inhibitor [116], suggesting that the NVJ may also play a role in inter-organelle ceramide trafficking. In fact, Mdm1 defines the site of LD budding in the interface between ER, vacuole, and LD by presumably channeling free FAs from locally enriched FA pools into LDs [117] (Figure 2D). Because FAs are converted with DAG into TAGs to be taken up by LDs and TAG synthesis is carried out be the same enzymes that mediate acylceramide synthesis, Mdm1 might be involved in acylceramide transport into LDs as well. Nvj2 and tricalbins are integral ER membrane proteins with a single TM domain and multiple TM domains, respectively [114], and they facilitate non-vesicular transfer of ceramide. Neither deletion of Nvj2 nor tricalbins can completely prevent IPC synthesis via the non-vesicular transport of ceramide [12,113]. They may have a parallel cooperative function in non-vesicular ceramide transport (Figure 3A). Otherwise, other proteins are required for the non-vesicular transport process. A study with a reconstituted cell-free system showed that non-vesicular ceramide transport involves heat-sensitive cytosolic protein(s) [31]. In mammalian cells, CERT is a well-described soluble transfer protein that delivers ceramide to the Golgi at the ER-Golgi contact sites [118,119]. Identifying soluble proteins that facilitate non-vesicular ceramide transfer in yeast is an important challenge for uncovering coordinated mechanisms.

## 6. MCSs and LDs in Ceramide Lipotoxicity

Perturbations in sphingolipid metabolism are linked to cellular dysfunction and cell death. In fact, in *S. cerevisiae*, inhibition of IPC synthase activity by aureobasidin A, a specific inhibitor of Aur1, leads to cell death or growth inhibition [121,122,123,124,125,126,127]. This is thought to be due to the accumulation of ceramides or the reduction of complex sphingolipids [123,124,127,128,129]. Because excess ceramide is toxic, ceramide levels in organelles are tightly controlled by synthesis, transport, and catabolism. Yeast has three mechanisms for preventing toxic accumulation of ceramide: converting it to complex sphingolipids after transferring it to the Golgi apparatus (Figure 1A and Figure 2B), converting it to acylceramide and storing it in the LDs (Figure 1A and Figure 2D), and hydrolyzing it into a free fatty acid and sphingoid base (Figure 1A). Because the deletion of Nvj2, Dga1, and Lro1 leads to a synthetic growth defect [113], ER-Golgi contact-mediated ceramide transport and acylceramide synthesis seems to be redundant in the clearance of toxic ceramides. Combinational mutations of tricalbins and Sec proteins that drive vesicular transport cause a dramatic accumulation of LDs [12], suggesting that vesicular transport from the ER is also important in preventing ceramide stress (Figure 3B). Given that excess ceramides are converted into acylceramides in the ER and acylceramides are incorporated into the LDs [10,11,12,113], ER-LD contacts may play a role in protection against ceramide toxicity. Consistent with this, a recent study showed that loss of ER-LD contacts results in increased levels of ceramides and their sphingoid precursors [130], even though its effect on acylceramide levels remains unclear. Interestingly, it was suggested that ER-LD contacts are one of the places where sphingoid intermediates are synthesized, and that seipin, an ER protein involved in LD biogenesis, negatively regulates sphingoid production at ER-LD contact sites [130] (Figure 2D). Thus, ER-LD contact sites may help prevent the accumulation of toxic ceramide not only by facilitating the formation of acylceramide and its incorporation into the LD, but also by reducing the levels of sphingoid intermediates. Degradation of ceramides is carried out by two ceramidases, which are integral membrane proteins localized to the ER and have different substrate specificities, such that Ypc1 preferentially hydrolyzes phytoceramide, whereas Ydc1 hydrolyzes dihydroceramide [131]. There is evidence that Ypc1 can rescue the growth defect caused by reduced Aur1 activity, probably by reducing toxic ceramide levels [127]; however, whether the ceramidase activity is regulated at ER-organelle contact sites remains to be investigated.

Ceramide toxicity depends on the hydroxylation state and acyl chain length. Phytoceramide formed by hydroxylation of dihydroceramide at C-4 seems to be less toxic than dihydroceramide [132]. By replacing endogenous ceramide synthases with one of the enzymes from cotton a yeast strain (GhLag1) was engineered that produces C18 rather than C26 ceramides. GhLag1 cells remain viable, even if *AUR1* is deleted, indicating that C18 ceramide is not toxic to yeast cells [129]. How long-chain ceramides drive yeast cell death is unclear, but ceramides were shown to form pores in the outer membrane of isolated mitochondria from mammalian and yeast cells [133,134,135] (Figure 3B). The formation of such ceramide channels can mediate the passage of cell death signals from mitochondria, such as cytochrome c [136], apoptosis-inducing factor [137], and endonuclease G [138]. ER-mitochondria contacts may facilitate ceramide-mediated mitochondrial apoptosis. Remarkably, ER-mitochondria encounter structure (ERMES) components are required for the acetic acid-induced yeast apoptosis associated with an increase in ceramide levels [136,139]. Similar to mammalian cells [140,141,142], yeast mitochondria can also form contacts with LD. Such mitchondria-LD contacts may alleviate ceramide toxicity by promoting the transfer of ceramide from the mitochondria to LD or serve as a site to deliver toxic ceramides to the mitochondria to trigger apoptosis. Consistently, LD encompasses ceramides in wild-type cells treated with aureobasidin A or in GhLag1. Most of the ceramides found in LDs isolated from GhLag1 cells are C24 ceramides, even though the most abundant cellular ceramide in the GhLag1 mutant is C18 ceramide, suggesting that excess toxic long-chain ceramides may be segregated in the ER and sorted with acylceramides to the LD [129]. Furthermore, ceramides are found in the vacuole [143], which forms contact sites with mitochondria, called vacuolar and mitochondrial patches (vCLAMP) [144,145], and with LD [117,146], in addition to contact sites with the nuclear ER, called NVJ [145,147]. These contact sites may also help to prevent toxic ceramide accumulation.

## 7. Conclusions and Outlook

Since J. L. W. Thudichum named the backbone of sphingolipids “sphingosin” in 1884 for its enigmatic (“sphinx-like”) properties, we have learned a lot about the structures, synthetic pathways, and functions of sphingolipids. Great strides in sphingolipid biology have been made in the past few decades, through studies using yeast. In addition to the identification of genes involved in sphingolipid synthesis and catabolism and the mechanisms of post-translational regulation of sphingolipid synthesis, through phosphorylation and ubiquitination, some tethers that physically bridge the pairs of organelles at the contact sites have recently been demonstrated to play a role in sphingolipid trafficking. However, we are only beginning to appreciate the potential of MCSs as important regulators of sphingolipid homeostasis. We still have a limited understanding of how cells sense the levels of sphingolipids and MCS-mediated processes are regulated and coordinated to maintain sphingolipid homeostasis. Furthermore, the molecular mechanisms of sphingolipid homeostasis at the transcriptional level and their upstream signaling cascades are poorly understood, although some enzymes involved in sphingolipid metabolism have been shown to be transcriptionally regulated [70,148,149] (Figure 1B).

Another issue is the toxicity caused by excess ceramide accumulation. Although there is evidence that a reduction in complex sphingolipid levels causes mitochondria-dependent apoptosis [124], direct evidence that excess endogenous ceramides trigger mitochondria-dependent cell death has not yet been shown in yeast. Defining the molecular basis of ceramide-induced yeast cell death will pave the way for a better understanding of the potential roles of MCSs in protection against ceramide toxicity. Certain cellular stresses, such as heat and ER stress, induce an increase in ceramides [85,129,150,151,152]. Bridging the gap in our understanding of how cells adapt to such stresses, with increased ceramide levels [113,151,152,153,154], will provide new insights into the physiological significance of sphingolipids and answer the fundamental questions of why cells use ceramide as a messenger of cell death and require two routes via vesicular and MCS-mediated non-vesicular pathways for ceramide trafficking.

Finally, cells must have mechanisms to sense the quality of membrane sphingolipid composition, which is coordinated by sterol molecules interacting with sphingolipids in the membrane domains [155,156,157]. As the supply of sterols to membranes and the membrane domain formation are regulated by the transport of sterols through the MCSs [31,158,159,160], the study of the role of the MCS in sterol homeostasis will provide additional clues to our understanding of the mechanisms of sphingolipid homeostasis.

## Figures and Tables

**Figure 1 membranes-11-00971-f001:**
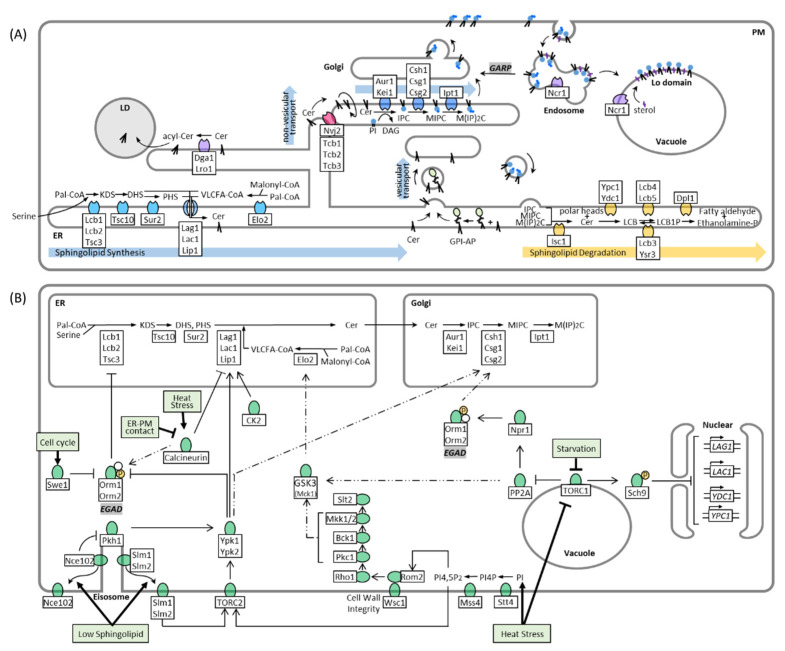
Sphingolipid synthesis, catabolism, and regulation of sphingolipid metabolism. See texts for details. (**A**) Sphingolipid biosynthesis and catabolism: sphingolipid biosynthesis begins at the ER and ceramide is produced in the ER, before it is transported to the Golgi apparatus. This figure does not depict all possible subclasses of ceramide that arise from hydroxylation, desaturation, and varied chain length. The transport of ceramide from the ER to the Golgi depends mainly on the COPII-mediated vesicular transport, and minorly non-vesicular transport via membrane contact sites (MCSs). Ceramide is converted to complex sphingolipids in the Golgi apparatus, and complex sphingolipids are sorted to the PM and endosome/vacuole by vesicular transport mechanisms. Isc1 cleaves complex sphingolipids, yielding ceramide and a free polar head group. Ceramide is hydrolyzed by ceramidases into LCB and fatty acid, and LCB is phosphorylated by LCB kinases to yield LCB1P, which is degraded by a LCB1P lyase to produce ethanolamine phosphate and fatty aldehyde [22,23]. (**B**) Regulation of sphingolipid metabolism: kinase and phosphatase signaling from the PM and other organelles regulate the various stages of sphingolipid metabolism. Regulation by proposed but insufficiently verified mechanisms are indicated by a dashed arrow. Abbreviations: palmitoyl-CoA (Pal-CoA), 3-ketosphinganine (KDS), dihydrosphingosine (DHS), phytosphingosine (PHS), long-chain base (LCB), long-chain base 1-phosphate (LCB1P), very long-chain fatty acid (VLCFA), ceramide (Cer), acylceramide (acly-Cer), lipid droplet (LD), GPI-anchored protein (GPI-AP), phosphatidylinositol (PI), diacylglycerol (DAG), phosphatidylinositol (IPC), mannosyl inositolphosphorylceramide (MIPC), mannosyl di(inositolphosphoryl) ceramide (M(IP)2C), ethanolamine phosphate (Ethanolamine-P), Golgi-associated retrograde protein (GARP), endosome and Golgi-associated degradation (EGAD), phosphatidylinositol-4-phosphate (PI4P), and phosphatidylinositol (4,5)-bisphosphate (PI4,5P_2_).

**Figure 2 membranes-11-00971-f002:**
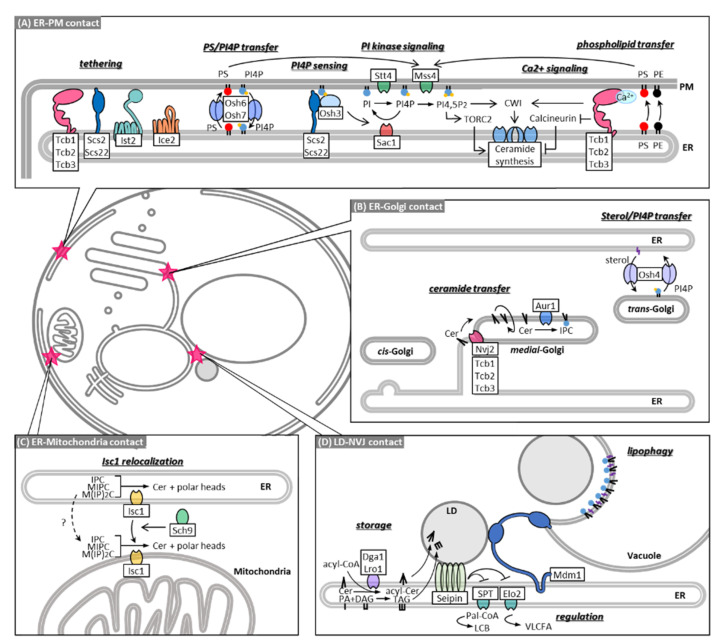
Membrane contact sites in sphingolipid metabolism. Various MCSs are involved in sphingolipid transport and homeostasis. See text for details. (**A**) ER-PM contact sites are tethered by tricalbins (Tcb1-3), Ist2, Scs2, Scs22, and Ice2. They are essential for Ca^2+^ signaling and maintaining lipid homeostasis, including PS and PI4P. (**B**) ER-medial Golgi MCS is proposed to be the site for non-vesicular transport of ceramide. (**C**) Isc1 localizes to the ER and the mitochondrial outer membrane. Sch9 is essential for the proper translocation of Isc1 from the ER to the mitochondria. (**D**) Acylceramide converted from ceramide is sorted into the LDs with TAG. Mdm1 acts in the interface between ER, vacuole and LD and spatially defines the site of LD budding. Abbreviations: cell wall integrity (CWI), phosphatidylserine (PS), phosphatidylethanolamine (PE), triacylglyceride (TAG). See Figure 1 for others.

**Figure 3 membranes-11-00971-f003:**
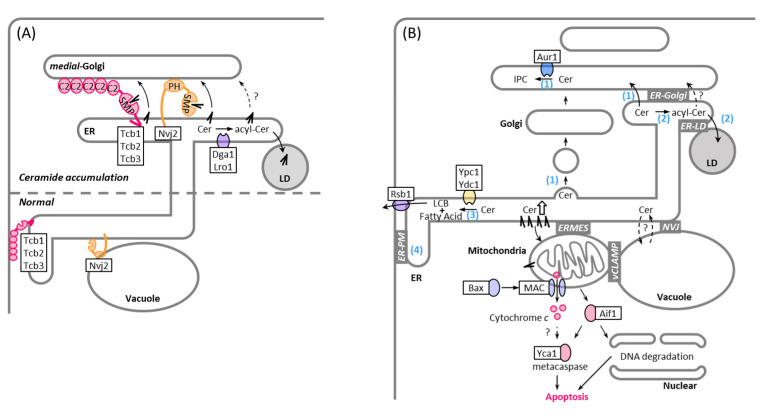
Membrane contact sites in non-vesicular transport and lipotoxicity of ceramide. See text for details. (**A**) Models of non-vesicular ceramide transport: non-vesicular ceramide transport is proposed to occur between the ER and medial Golgi. Under non-stress conditions, tethering protein Nvj2 is mainly localized at the NVJ and tricalbins are localized to the ER, where they form contacts with the PM (lower part). They re-localize to ER-medial Golgi contacts, following ER stress or ceramide overproduction (upper part). SMP domains of Nvj2 and tricalbins likely transfer ceramide from the ER to the Golgi. (**B**) Mechanical scheme of ceramide-mediated apoptosis and pathways of ceramide removal from the ER: ceramide transfer to the Golgi and conversion into IPC (1), ceramide conversion into acylceramide and storage in the LDs (2), ceramide hydrolyzation into LCB and free fatty acid (3), and LCB transport from the cytoplasmic side toward the extracytoplasmic side [120], probably at ER-PM contact site (4). Abbreviations: mitochondrial apoptosis-induced channel (MAC), nucleus-vacuole junction (NVJ), vacuole and mitochondria patch (vCLAMP), endoplasmic reticulum-mitochondria encounter structure (ERMES). See Figure 1 and Figure 2 for others.

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
