# Peer review of "Membrane Contact Sites in Yeast: Control Hubs of Sphingolipid Homeostasis"

_membranes, 2021, doi:10.3390/membranes11120971_

Round 1

Reviewer 1 Report

Sphingolipids are not only the structure components of membrane, but also act as intracellular signaling molecules in various processes, including apoptosis. In this manuscript, Schlarmann et al., reviewed the role of membrane contact sites (MCSs) in sphingolipid metabolism and ceramide transport and the mechanisms that prevent toxic ceramide accumulation in budding yeast Saccharomyces cerevisiae. This is a very nice review. The authors have fully discussed the synthesis, catabolism, and trafficking of sphingolipid in yeast, the regulation mechanism of sphingolipid metabolism, the roles of MCSs in sphingolipid metabolism, the roles of MCSs in non-vesicular transport of ceramide, and ceramide lipotoxicity. I only have few concerns.

  1. The title is not quite adequacy.
  2. Some sentences are very hard to read due to too long.
  3. Sphingolipids and sterols are interactional and interdependent. Although the mainly topic of this review is the role of MCSs in sphingolipid metabolism and ceramide transport. It is better to outlook the role of MCSs in sterols in the last section.

Author Response

Reviewer #1

Comments to the Author:

Sphingolipids are not only the structure components of membrane, but also act as intracellular signaling molecules in various processes, including apoptosis. In this manuscript, Schlarmann et al., reviewed the role of membrane contact sites (MCSs) in sphingolipid metabolism and ceramide transport and the mechanisms that prevent toxic ceramide accumulation in budding yeast Saccharomyces cerevisiae. This is a very nice review. The authors have fully discussed the synthesis, catabolism, and trafficking of sphingolipid in yeast, the regulation mechanism of sphingolipid metabolism, the roles of MCSs in sphingolipid metabolism, the roles of MCSs in non-vesicular transport of ceramide, and ceramide lipotoxicity. I only have few concerns.

We thank the reviewer for the positive appreciation of our manuscript and very useful suggestions.

1) The title is not quite adequacy.

We changed the title like this: “Membrane contact sites in yeast: Control hubs of sphingolipid homeostasis”

2) Some sentences are very hard to read due to too long.

We shortened most of the long sentences in the new version.

3) Sphingolipids and sterols are interactional and interdependent. Although the mainly topic of this review is the role of MCSs in sphingolipid metabolism and ceramide transport. It is better to outlook the role of MCSs in sterols in the last section.

We mentioned about the role of MCS in sterols in the last section (7. Conclusion and outlook) and cited the related papers.

Reviewer 2 Report

This review is very well written and expanded basic reviews about sphingolipids . It is of interest for many readers.

Only one criticism is that fig.1 is too complicated and hardly readable 

Author Response

Reviewer #2

Comments to the Author:

This review is very well written and expanded basic reviews about sphingolipids. It is of interest for many readers.

We thank the reviewer for the positive appreciation of our manuscript.

1) Only one criticism is that fig.1 is too complicated and hardly readable.

To make the Figure 1 simpler, we divided it to (A) Sphingolipid biosynthesis and catabolism and (B) Regulation of sphingolipid metabolism. 

Reviewer 3 Report

The “Sphingolipids and membrane contact sites in yeast” article is a very nicely written and comprehensive review of a topical subject. As such, it makes an important contribution to the field and I have only minor suggestions for improvement. I did not see any major flaws and would not expect the authors to respond to every comment below. Hopefully, they will find these comments useful while revising the manuscript.

  1. Page 3 Line 88 Shouldn't ceramide synthesis should be ceramide synthase?
  2. Page 5 Line 124, "GPI anchored proteins are also synthesized in the ER and transported through the Golgi apparatus to the PM surface by COPII vesicles."  Instead of COPII vesicle, I would use vesicular transport as COPII mediates transport between ER to Golgi.
  3.  Although mostly authors discussed about yeast-related data. It would have really benefited the community to have a broader perspective to have some discussion about mammalian cell line data. 
  4.  I think Ceramide transport protein (CERT) could have been discussed briefly in the non-vesicular transport section. 

Author Response

Reviewer #3

Comments to the Author:

The “Sphingolipids and membrane contact sites in yeast” article is a very nicely written and comprehensive review of a topical subject. As such, it makes an important contribution to the field and I have only minor suggestions for improvement. I did not see any major flaws and would not expect the authors to respond to every comment below. Hopefully, they will find these comments useful while revising the manuscript.

We thank the reviewer for the positive appreciation of our manuscript and very useful suggestions.

1) Page 3 Line 88 Shouldn't ceramide synthesis should be ceramide synthase?

Thank you very much. We corrected this in the new version.

2) Page 5 Line 124, "GPI anchored proteins are also synthesized in the ER and transported through the Golgi apparatus to the PM surface by COPII vesicles."  Instead of COPII vesicle, I would use vesicular transport as COPII mediates transport between ER to Golgi.

Thank you very much. We corrected this in the new version.

3) Although mostly authors discussed about yeast-related data. It would have really benefited the community to have a broader perspective to have some discussion about mammalian cell line data. I think Ceramide transport protein (CERT) could have been discussed briefly in the non-vesicular transport section.

We have already referred to mammalian cell lines in some cases where data from yeast is missing (e.g “calcineurin, which promote endocytosis universally in neuronal and non-neuronal secretory cells [104]”). I have therefore added this to the introduction: “We have focused on studies using S. cerevisiae but also highlight research on mammalian cell lines when complementation is needed.” We also briefly mentioned about CERT in the non-vesicular transport section (5. MCSs in non-vesicular transport of ceramide) and cited the related papers.